# The distribution of functional N-cycle related genes and ammonia and nitrate nitrogen in soil profiles fertilized with mineral and organic N fertilizer

**Massimo Zilio**[1], **Silvia Motta**[2], **Fulvia Tambone**[1], **Barbara Scaglia**[1], **Gabriele Boccasile**[3], **Andrea Squartini**[4], **Fabrizio Adani**[1] *

**1** Gruppo Ricicla labs., DiSAA, Università degli Studi di Milano, Milan, Italy, **2** Ente Regionale per i Servizi alla Agricoltura e alle Foreste, Regione Lombardia, Milan, Italy, **3** DG Agricoltura, Regione Lombardia, Milan, Italy, **4** DAFNAE, Università degli Studi di Padova, Agripolis, Padua, Italy

* fabrizio.adani@unimi.it

**Data Availability Statement:** All relevant data are within the manuscript and its Supporting Information files.

## Abstract

Nitrogen transformation in soil is a complex process and the soil microbial population can regulate the potential for N mineralization, nitrification and denitrification. Here we show that agricultural soils under standard agricultural N-management are consistently characterized by a high presence of gene copies for some of the key biological activities related to the N-cycle. This led to a strong $NO_3^-$ reduction (75%) passing from the soil surface (15.38 ± 11.36 g $N$-$NO_3$ $kg^{-1}$ on average) to the 1 m deep layer (3.92 ± 4.42 g $N$-$NO_3$ $kg^{-1}$ on average), and ensured low nitrate presence in the deepest layer. Under these circumstances the other soil properties play a minor role in reducing soil nitrate presence in soil. However, with excessive N fertilization, the abundance of bacterial gene copies is not sufficient to explain N leaching in soil and other factors, i.e. soil texture and rainfall, become more important in controlling these aspects.

## Introduction

Anthropogenic activities are the major driver of changes in the global nitrogen (N) cycle since the last century, resulting in N-flows being 3.3-fold higher than those due to natural processes, achieving a total globally fixed nitrogen of 413 Tg N $y^{-1}$. Since nitrogen is one of the most important nutrients for many life forms, such a strong change in its availability has important effects on the balance of terrestrial and aquatic ecosystems [1,2].

Agriculture has indeed a major role in this process: the global amount of N used in agriculture has increased from 12 Tg N in 1960 to 104 Tg N in 2010 and the amount of $N_2$ fixed to $NH_4^+$ by industrial processes and destined for agriculture contributes today to 45% of the total nitrogen fixed annually on the planet [1,3,4].

As a consequence of that, the total amount of N brought to the soil today, on a world scale, is more than twice that considered to be within a safe planetary boundary, i.e. a safe operating

**Funding:** Grant recieved by Fabrizio Adani -University of Milan CONVENZIONE QUADRO Regione Lombardia – ERSAF, Italy: Supporto tecnico per l'applicazione e il monitoraggio della direttiva nitrati (ARMOSA) + rapporto ambientale per VAS, piano triennale 2014 – 2016, Agreement Collaborazione tecnico-scientifica per azioni finalizzate a valutare la sostenibilità complessiva della gestione". Project N. 15-3-3014000-218 and 15-3-3014000-219. Gruppo Ricicla lab. Università degli Studi di Milano, Italy, Project N. 1705, 2011: RV_ATT_COM16FADAN_M.

**Competing interests:** he authors have declared that no competing interests exist.

space for humanity to avoid the risk of ecosystems' deterioration [5–7]. Furthermore, an increase in the use of N in the food system by 51% in 2050 has been estimated, with a global increase in the environmental pressure of the food system estimated to be in the range 52%-90%, in the absence of mitigation measures [7,8]. Without emission reductions, global N losses are expected to further increase, reaching in 2050 levels equal to 150% of those of 2010 [9].

Nitrogen fertilizers applied to agricultural soils cause the release of N into the environment, both in the atmosphere ($NH_3$, $N_2O$, $N_2$) and in groundwater ($NO_3^-$) [10]. In particular, the mechanisms that regulate the amount of nitrate ($NO_3^-$) leached by the soil over time have been studied for many years: N concentration in soil, total N dosed to the soil, climate and rainfall, and soil texture have all been reported to affect nitrate leaching.

Higher nitrate concentration leads to higher nitrate leaching depending on the amount of rainwater that percolates into the soil in a given period of time [11]. However, the amount of rainwater arriving in the soil depends upon the climate. In temperate geographical areas the period between autumn and early spring that is characterized by intense rain events and low N uptake by plants, is considered to be at greater risk of nitrate leaching (15–19% of the total N dosed). The leaching risk is reduced during the summer season (8–11% of the total dosed N) [12].

Water movement in soil depends upon soil texture that assumes an important role in regulating nitrate leaching. Generally, sandy soils are more at risk of leaching than silty or clayey soils, with a leaching loss ratio of 5:1 for a silt loam soil compared to a clay loam soil [13].

For agricultural soils both nitrate concentration and leaching depend in turn on the amount of N that is supplied to the soil with fertilizer. Following this, the portion of N not absorbed by the plants is converted into nitrate through the nitrification process, which involves the oxidation of ammonium carried out by nitrifying microorganisms [13]. Sebilo et al. [14], using radio labelled N ($\delta^{15}N$), have shown that approximately 37% of the nitrogen dosed to the soil was not immediately taken up by plants, but became part of the organic nitrogen pool of the soil, and after 30 years, 15% was still part of it. This nitrogen is slowly mineralized by soil microorganisms and leached even many years after its application [14].

Nitrate that is not leached and remains in the soil can be metabolized and transformed by the resident microbial communities by the denitrification process [3,13,15–18], returning to the atmosphere in gaseous form ($N_2O$, $N_2$). Therefore, biological processes seem to play an important role in determining nitrate presence and leaching in soils. Today there is general agreement that soil microorganisms play a central role in the N cycle [3,13,17] as they are responsible for the conversion of N in its various forms: the structure of the soil microbial populations is regarded as the major variable that regulates the potential for N fixation, mineralization, nitrification and denitrification [19].

In recent years, some findings have enriched the complexity of the known pathways ruling the nitrogen cycling in the environment. One of such cases is the comammox [20], in which one single species of the genus Nitrospira has been demonstrated to perform a complete phenotype of nitrification from ammonium to nitrate. Another previously overlooked distinction is the split of the nitrous oxide reductase genotype in two clades (*nosZ* I and *nosZ* II) the second of which corresponds to novel non-denitrifying types of $N_2O$ reducers [21].

Despite the existence of genetic variants linked to the N cycle-related genes in soil, we have chosen a subset of genes as suitable proxies for nitrification, denitrification and nitrogen fixation: archaeal *amoA*, eubacterial *amoA*, *nirK*, *nosZ* and *nifH*. The choice was made based on the recent literature reporting useful considerations that seem to confirm our choice.

With respect to the comammox *Nitrospira*, which was originally discovered upon enriching cultures from material found in a biofilm growing within a steel pipe in deep oil wells [20], subsequent studies addressed the relevance of such taxa in agricultural contexts [22] and

concluded that although the species is present in soils, the dominant contributors of potential nitrification are the classic ammonia oxidizing bacteria and the newly discovered comammox do not play a significant role in these pathways ($P < 0.05$).

Concerning the second *nosZ* clade (*nosZ* II) we chose to restrict the survey of the terminal gene of denitrification to *nosZ* I based on the conclusions of Domeignoz-Horta and coworkers (2015) [23] who reported that: (a) the *nosZ* I community was consistently more abundant than the *nosZ* II one and (b) no significant differences between the two groups could be ascribed to the different agricultural management practices, either in relation to crops or to fertilization regimes. The same authors add that the lack of detectable variations between these subgroups is in line with the fact that such differences were reported only in long-term agronomic trials that had been carried out for over 50 years.

Finally, regarding the known existence of two families of genes able to perform nitrite reductase activity, converting nitrite into nitrous oxide (*nirK* and *nirS*), we selected the former for the following reasons: (a) *nirK*-harboring bacteria mostly dominate in soils and rhizospheres over nitrite reducers of the nirS kind [24]; (b) there is a tight correlation between *nirS* and *nosZ* [25] which allows us to infer indirect information on the abundance of the former by analyzing the latter. These considerations are also confirmed by our prior work [26] in which we analyzed both *nirK* and *nirS* as well as *nosZ* in Bermuda grass rhizospheres.

Enhanced N removal through optimization of denitrification has drawn much attention as an effective approach towards N control because it is the only pathway, except for the process of anaerobic ammonium oxidation (Anammox), by which reactive forms of nitrogen (Nr) in terrestrial and aquatic ecosystems are transformed back into inert $N_2$ gas [15,27]. When not converted into gaseous forms, N stored in soil can be progressively leached as $NO_3^-$ polluting groundwater and shallow water bodies [14,28,29]. Understanding the dynamics of N transformation and movement in soils is complex because of the large number of variables affecting this process [14,15,30–33].

Nitrogen fertilization and the type of N fertilizers have also been reported to affect N-related microbial populations [34]. Indeed, a drastic effect on the balance ammonia oxidizing archaea (AOA) vs. eubacteria (AOB) was observed, with the AOBs stimulated by the providing of N to the soil, especially organic N, while the AOBs seemed to be less responsive, or even inhibited by both mineral and organic fertilization [35]. For denitrifying bacterial populations, a generic increase in their number was reported in long-time N-fertilized soils compared to unfertilized, especially in the case of organic fertilizer use (manures). On the other hand, fertilizations with sewage sludge showed a negative effect on these microorganisms, probably due to the acid pH of this type of fertilizer [36].

In addition, environmental factors, such as pH, soil texture, humidity and nitrogen availability, are reported to affect the structure and size of soil microbial populations, though the interdependency of these factors with microbial populations in relation to the N-cycle is still largely unclear [17,37–42].

This study aims at determining the potential of agricultural soils to reduce nitrate concentration down to a one-meter depth, focusing on the role of soil microorganisms related to the N-cycle in transforming reactive N. To do this, we investigated reactive forms of nitrogen distributed along the profiles (0–25 cm, 25–50 cm, 50–75 cm and 75–100 cm) of twelve different agricultural soils located in the Po Valley (northern Italy), one of the most intensive agricultural areas of the EU, managed with different N-fertilization for both total N dosed and N-fertilizer types, during three growing seasons. The N cycle-related microbial communities were PCR-quantified down the soil profiles [43], and data collected from 308 samples were critically compared with chemical data, i.e. ammonia N-$NH_4$ and nitrate (N-$NO_3$), and soil properties to draw a clear picture of the potential of soil in reducing nitrate-leaching.

## Material and methods

### Experimental sites

Twelve soil sites cultivated with cereals (mainly corn) and distributed in eight different localities in the Po valley (Italy) were considered in the years 2014, 2015, 2016. These soils were chosen because they belong to farms of the nitrate-monitoring network of the Lombardy Region (ARMOSA Network–EU—Nitrate Directive 676/91 CEE, Regione Lombardia–ERSAF, Italy).

Ten of them: soil codes 1a, 1b, 2, 3a, 3b, 4a, 4b, 5, 6a, 6b, were fertilized by a regular farming approach using different types and quantities of nitrogen up to a maximum of 450 kg N ha$^{-1}$ (Stage 1 of the study). The last two soils, soil codes 7 and 8, received in 2016 an excess of N fertilizers (Stage 2 of the study). In particular, Soil 7 was equivalent to Soil 4a but it received an extra N-fertilization in October (860 kg N Ha$^{-1}$, for a total annual N of 1,243 kg N Ha$^{-1}$) by using pig slurry (S1 Table). Soil 8 represented a field cropped with maize and received, during the season, an excess of N (1,470 kg N Ha$^{-1}$) by three N-fertilization events during the year, using digestate in April, urea at the end of July and pig slurry in October (S1 Table).

Soils with the same code number but different letters were carried out at the same site (farm), but in different fields and with different N fertilizers. The agronomic management of each site is reported in S1 Table.

### Sampling

For each experiment, between March 2014-October 2016, soil cores at 4 depths (0–25 cm; 25–50 cm; 50–75 cm; 75–100 cm) were collected. At least 5 samples were taken for each experiment during the crop cycle, in correspondence with the principal phenological plant stages. Sampling periods are reported in S2 Table.

For each soil sampling a composite sample was taken, formed by mixing 10 sub-samples taken inside the plot. The collection points within each experimental plot were identified according to an X distribution, taking care to avoid the borders of the plots.

Soils taken for chemical analyses were put into sealed containers and stored at 4˚C; analyses were performed starting the next day. Soil samples for DNA extraction and qPCR were processed in the hours immediately following the sampling.

### DNA extraction and target gene quantification by qPCR

Total DNA extractions were carried out on a quantity of soil equal to 5g per sample. For total DNA extraction, the NucleoSpin® Soil (Macherey-Nagel) kit was used. The total DNA extracted was then quantified and normalized using the Quant-iT ™ PicoGreen ™ dsDNA Assay Kit (Thermo Fisher Scientific). Real time PCR reactions were performed on real-time 7900HT (Applied Biosystems) using SyberGreen technology, in a final volume of 10μl. The sequences of the primers used are reported in S3 Table. Each sample was tested in triplicate, and the standard calibration curve was built using five points in triplicate, equal to fifteen reactions. As templates for the standard curves, amplicons for each of the target genes were cloned into purified plasmids (pGem-T; Promega Corp.) and inserted into *E. coli* JM101 by electroporation. Knowing the size of the vector (3,015 bp) and those for each insert (data from literature, S3 Table), and measuring the plasmid DNA concentration, the number of copies per ng of DNA and the corresponding amounts to be used for each of the quantitative PCR calibration curves, were calculated. The number of gene copies per gram of soil in each sample was then calculated by comparing the output of the qPCR reaction with the calibration curve for the corresponding gene. Data analysis was performed using SDS v2.1 software (Applied Biosystems).

## Soil analysis

The soil cores were stored at 4°C after collection and analyzed. In particular the soil ammonia (N-$NH_4$) and nitrate (N-$NO_3$) concentration were determined through Kjeldahl distillation using Devarda's alloy [44]. The soil phosphorus content ($P_2O_5$) was determined by extraction in bicarbonate [45]. Organic carbon content was determined by the dichromate method [44], cation exchange capacity (CEC) by saturating the samples with $BaCl_2$-triethanolamine solution (pH 8.1) [46], and soil pH in aqueous solution using a 1:2.5 sample/ water ratio [47]. Soil texture was determined by the pipette method [48]. All the analyses were performed in triplicate. The values shown represent the average of the three replicates, with a standard deviation always below 5%.

## Statistical analysis

Principal Component Analysis (PCA) was performed using the excel package Addinsoft XLSTAT™. For two-way ANOVA analysis and linear correlation analysis the IBM SPSS™ Statistics software was used, with a significance threshold set at 0.05. Multiple comparisons were performed with the Duncan and Gabriel methods. When necessary, datasets were normalized as $log_{10}x$, and homogeneity of variances was tested through the Levene test.

Multiple linear regressions for nitrate content in soils (75–100 cm) ($n = 10$) vs. agronomic, chemical, physical, biological and meteorological data (parameters = 26) were done using the partial least square method (PLS). The cross-validation leave-one-out approach of un-scaled variables was applied to calculate the goodness of regressions (goodness of fit coefficient-$R^2$ and goodness of prediction coefficient- $R^2cv$, respectively). Taking into consideration all variable values, the PLS regression was calculated and the importance of each independent variable (importance coefficient) defined. Then PLS analysis was repeated removing those variables characterized by the coefficient of the least importance. This procedure was repeated until a final regression model with goodness of regressions coefficient ($R^2$ and $R^2cv$) and the smallest number of variables was achieved. PLS was performed using SCAN software (Minitab Inc., State College, PA).

# Results

## Nitrogen concentration in agricultural soil

We started by concentrating our work on the analysis of reactive nitrogen content, i.e. ammonium ($NH_4^+$) and nitrate ($NO_3^-$), in soil with normal N fertilization (Soils 1–6) (S1 Table) taken every 25 cm in depth, starting from the surface (0 cm) to 1-meter-deep, in the years 2014, 2015 and 2016. Measurements made ($n = 248$) (S4 Table) indicate that despite the soils differing for characteristics (Table 1) and management, with particular reference to N dosed (range of 153–453 kg $Ha^{-1}$ $y^{-1}$) and N-fertilizers used (urea, animal slurries and digestates from animal slurries) (S1 Table), the $NO_3^-$ concentrations (Fig 1) along soil profiles and during agricultural seasons showed similar trends by decreasing dramatically (reduction of 75% ± 14%; $n = 122$) proceeding downwards from the soil surface (average of 15.4 ± 11.4 mg N-$NO_3^-$ $kg^{-1}$; $n = 63$) to 1 m depth (average of 3.9 ± 4.4 mg N-$NO_3^-$ $kg^{-1}$; $n = 59$). Data measured on the surface are similar to those previously reported for agricultural soil [49] but surprisingly, those measured for 75–100 cm depth layer are comparable to those reported for natural soil [50]. Ammonium concentration (S1A Fig) shows a similar trend, reducing, on average, its content from 4.1 ± 9.6 mg N-$NH_4^+$ $kg^{-1}$ ($n = 63$) at the surface (0–25 cm) to 1.6 ± 2.7 mg N-$NH_4^+$ $kg^{-1}$ ($n = 57$) at 75–100 cm depth.

During the agricultural season, $NO_3^-$ content in soils' surface layers (0–25 cm) varied considerably depending upon: proximity to the fertilization event, total N-dose, presence of crop

**Table 1. Soil characteristics.** Texture (sand tot., silt tot. and clay) and pH (H$_2$O) for the four soil depths studied. USDA soil taxonomy, percentage of organic carbon (OC %) and cation exchange capacity (CEC) for the 0–25 cm layer.

| Soil Code | Depth (cm) | Soil type | SAND$_{tot}$ (%) | SILT$_{tot}$ (%) | Clay (%) | pH | OC% | CEC |
|---|---|---|---|---|---|---|---|---|
| **1** | 0–25 | Calcisols | 15.35 | 54.7 | 29.9 | 8.5 | 1.00 | 31.4 |
| | 25–50 | | 15.6 | 50.9 | 33.5 | 8.33 | | |
| | 50–75 | | 12.2 | 50.3 | 37.6 | 8.51 | | |
| | 75–100 | | 74.3 | 16.5 | 9.2 | 8.6 | | |
| **2** | 0–25 | Cambisols | 46.2 | 32.1 | 21.7 | 8.3 | 1.97 | 15.28 |
| | 25–50 | | 23.2 | 59 | 17.8 | 8.7 | | |
| | 50–75 | | 22.7 | 60.8 | 16.5 | 8.6 | | |
| | 75–100 | | 88.7 | 7 | 4.3 | 8.7 | | |
| **3–4–7** | 0–25 | Cambisol aquic | 48 | 30 | 22 | 6.2 | 1.60 | 14 |
| | 25–50 | | 60 | 10 | 30 | 7.1 | | |
| | 50–75 | | 60 | 10 | 30 | 7.1 | | |
| | 75–100 | | 60 | 10 | 30 | 7.1 | | |
| **5** | 0–25 | Cambisol aquic | 41.9 | 49.7 | 8.4 | 7 | 1.55 | 13.4 |
| | 25–50 | | 41.9 | 46.7 | 11.4 | 7.4 | | |
| | 50–75 | | 58.9 | 22.7 | 18.4 | 7.3 | | |
| | 75–100 | | 85.9 | 6.7 | 7.4 | 7.3 | | |
| **6** | 0–25 | Luvisols | 29 | 40.5 | 30.5 | 8.5 | 1.50 | 17.1 |
| | 25–50 | | 25.5 | 44 | 30 | 8.4 | | |
| | 50–75 | | 38 | 41 | 20.5 | 8.6 | | |
| | 75–100 | | 46.5 | 36.5 | 17 | 8.5 | | |
| **8** | 0–25 | Gleisols mollic | 38.9 | 43.7 | 17.4 | 7.9 | 2.70 | 23.1 |
| | 25–50 | | 41.9 | 39.7 | 19.4 | 8.3 | | |
| | 50–75 | | 52.9 | 29.7 | 17.4 | 8.4 | | |
| | 75–100 | | 73.9 | 15.7 | 10.4 | 8.6 | | |

and sampling date. This trend was not confirmed for the deeper soil layers analysed, that showed much lower variability during the year (Fig 1).

Soil receiving an excess of N fertilization (Soils 7 and 8) showed different and more variable patterns in terms of mineral N presence in soil that increased a lot in correspondence with the substantial N fertilization made (Fig 2). In particular, Soil 7 (Fig 2A) showed in autumn (October-November), in correspondence with the high N-fertilization received with pig slurry (860 kg N Ha$^{-1}$) (S1 Table), a presence of NO$_3^-$ in the surface layer (0–25 cm) (110.3 mg N-NO$_3^-$ kg$^{-1}$ ± 33.6; $n$ = 6), that was much higher than those reported for the same layer for soils ordinarily managed (Soils 1–6). Despite this, only a small part of the nitrate was detected at 75–100 cm depth in soil, the NO$_3^-$ concentration at the same depth in autumn being 6.85 mg N-NO$_3^-$ kg$^{-1}$ ± 1.91 ($n$ = 6). This value is much lower than that measured in the month of June (15.81 mg N-NO$_3^-$ kg$^{-1}$) after normal N fertilization with urea (138 kg N Ha$^{-1}$) and is in line with data reported for soil under ordinary management (Soils 1–6). In this case, differences in nitrate concentration depended on rainfall, which was double in June in comparison with that for October-November (Table 2). This result indicates that soil water content in this case affected nitrate leaching, in fact for that month rainfall registered was extraordinary, i.e. 266 mm, to be compared with average rainfall (2014–2019) of 75 ± 26 mm (Table 2). This anomaly is confirmed by the absence of any correlation between nitrate presence at 1 m depth and rainfall measured for Soil 7 during the trial ($p$ = 0.75; $n$ = 8).

Soil 8 instead, showed NO$_3^-$ concentrations at the surface which were much higher than those measured for soils fertilized at the normal rate (Soils 1–6), in particular after high N-fertilization (June and August) (620 kg N Ha$^{-1}$) (Fig 2B). Nitrate concentration at 1-meter depth in

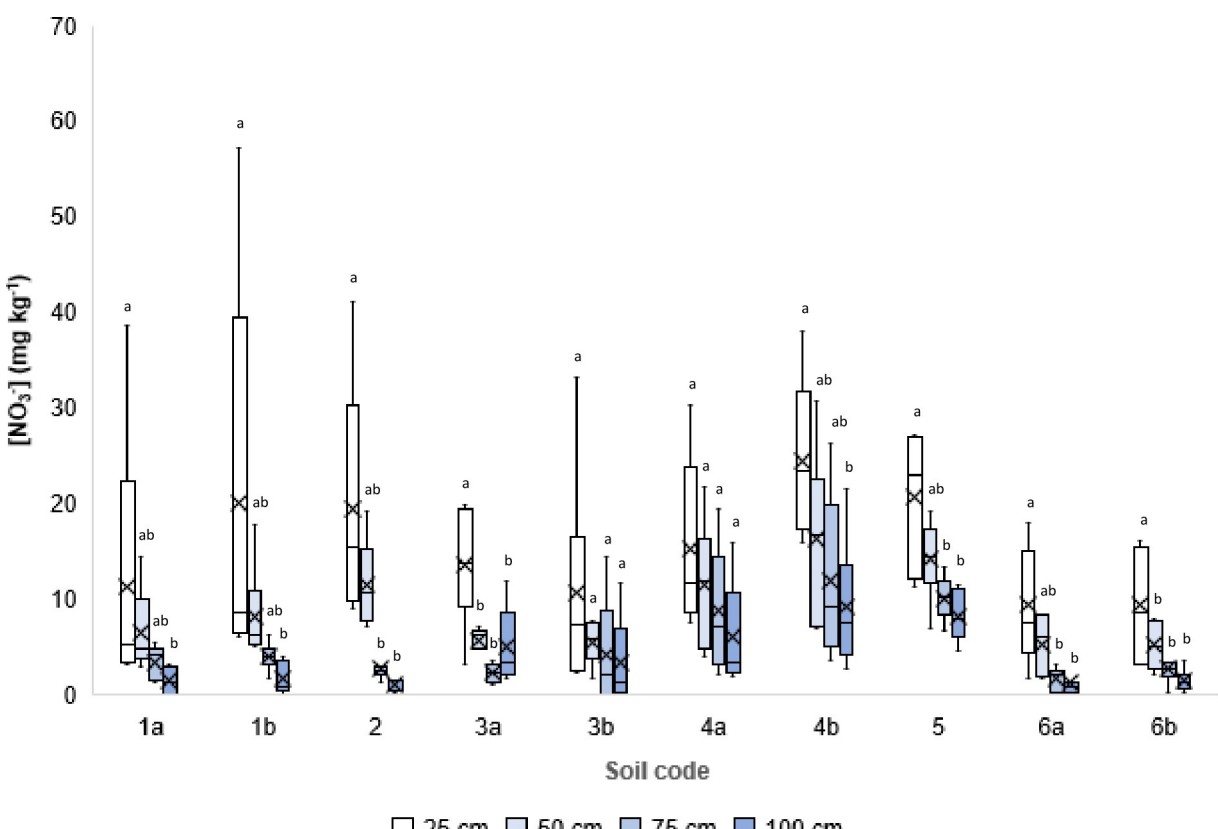

**Fig 1. Nitrate concentration in soil.** For each experiment, divided for depth classes, the box plot shows minimum and maximum values sampled (bars), the first and the third quartile (boxes), the median (lines inside boxes) and the average (crosses), $n = 248$. Letters within the same soil refer to One-way ANOVA tests (p<0.05, Gabriel post-test) performed for each soil.

this period was of 32.37 mg N-NO$_3^-$ kg$^{-1}$ ± 23.77 ($n = 3$) and in June, the NO$_3^-$ content exceeded 50 mg N-NO$_3^-$ kg$^{-1}$ at 1-meter depth, that is five times higher than values reported (on average) for the soils previously studied, including Soil 7 that was fertilized with an excess of N (total N supplied of 1,243 kg N Ha$^{-1}$) similarly to Soil 8 (total N supplied of 1,470 kg Ha$^{-1}$).

## The abundance of gene copies related to the N-cycle

Results for the quantification of DNA gene copies (gene copies g$^{-1}$ soil) coding for enzymes in charge of the N cycle, distributed along the profiles of the analysed soils (Soils 1–6) ($n = 252$) are reported in Fig 3 and S5 Table. It was interesting that the numbers of gene copies coding for different enzymes present at different soil depths were well correlated with each other ($0.57 < r < 0.90$, $p < 0.01$, $n = 252$) (S6 Table) above all if the first 50 cm were considered ($0.85 < r < 0.99$, $p < 0.05$, $n = 252$) (S7 Table). These data seem to suggest that nitrification, denitrification and N-fixation are interlinked [22].

Moreover, gene copies found per gram of soil decreased with depth for all soils studied (except for the gene *nifH*, related to N fixation), above all if 0–50 cm and 50–100 cm data were grouped (Fig 4). Consequently, we found that the number of genes copies (abundance) for the N cycle related genes were much higher in the surface layers (0–50 cm) than in the other layers (50–100 cm) (Fig 4). The difference observed was stronger in the case of bacterial *amoA* genes, i.e. the number of gene copies (gene copies g$^{-1}$ soil) for the layer 75–100 cm, were 16 times lower than those detected in the 0–25 cm layer.

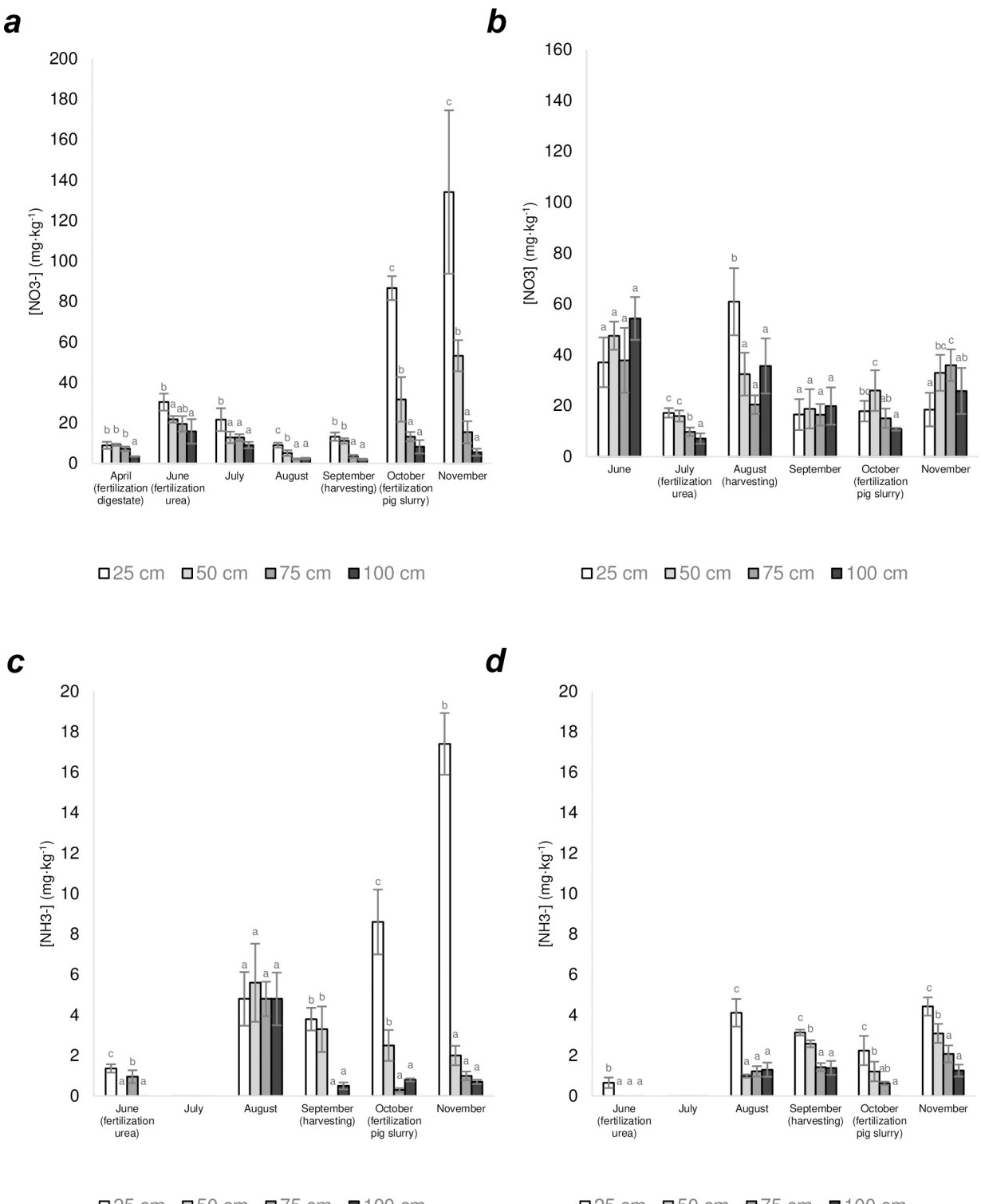

**Fig 2.** Nitrate (a and b) and ammonium (c and d) concentration in soil of the experiments 7 (a and c) and 8 (b and d) during the year 2016. Experiment 7 was supplied with a total amount of nitrogen of 1,423 kg N Ha$^{-1}$. Experiment 8 was supplied with a total amount of nitrogen of 1,470 kg N Ha$^{-1}$. Error bars show Standard Deviation ($n = 3$). Letters within the same soil refer to One-way ANOVA tests ($p < 0.05$, Gabriel post-test) performed for each soil.

**Table 2. Total rainfall for experimental sites studied (Soils 1–8).**

| MONTH | SOIL CODE | | | | | | | |
|---|---|---|---|---|---|---|---|---|
| | **1** | **2** | **3** | **4** | **5** | **6** | **7** | **8** |
| JAN | 152 | 225 | 52 | 34 | 22 | 30 | 34 | 26 |
| FEB | 136 | 137 | 127 | 178 | 121 | 88 | 178 | 116 |
| MAR | 31 | 56 | 25 | 45 | 68 | 90 | 45 | 43 |
| APR | 159 | 105 | 54 | 35 | 45 | 74 | 35 | 45 |
| MAY | 89 | 44 | 65 | 184 | 176 | 86 | 184 | 163 |
| JUN | 31 | 111 | 79 | 266 | 121 | 123 | 266 | 143 |
| JUL | 100 | 103 | 12 | 171 | 144 | 14 | 171 | 93 |
| AUG | 74 | 231 | 70 | 57 | 67 | 49 | 57 | 53 |
| SEP | 97 | 2 | 78 | 66 | 34 | 32 | 66 | 49 |
| OCT | 82 | 33 | 124 | 82 | 79 | 62 | 82 | 68 |
| NOV | 100 | 301 | 11 | 123 | 153 | 45 | 123 | 56 |
| DEC | 79 | 84 | 3 | 2 | 7 | 3 | 2 | 3 |
| TOTAL[a] | 1130 | 1432 | 700 | 1243 | 1037 | 696 | 1243 | 858 |
| ANNUAL AVERAGE[b] | 764 ± 232 | 911 ± 310 | 816 ± 229 | 816 ± 229 | 946 ± 190 | 575 ± 168 | 816 ± 229 | 903 ± 302 |

[t]Total rainfall (mm year$^{-1}$) recorded for the experiments site (Soil 1–8); data from ARPA Lombardia.

[b]Average rainfall for the period: 2014–2019 (mm year$^{-1}$) recorded for the experiments site (Soil 1–8)(mm, mean ± SD, $n$ = 6); data from ARPA Lombardia.

Soils 7 and 8 that received an excess of N fertilization (S1 Table) showed different patterns in comparison with Soils 1–6. In Soil 7 the number of gene copies coding for different enzymes correlated each other, as reported previously for Soils 1–6, although correlation coefficients were much lower ($0.4 < r < 0.93$, $p < 0.01$, $n$ = 32) (S8 Table). A different situation was observed in Soil 8. In this soil the gene copy numbers for the two genes responsible for the nitrification process (*amoA*-Archaea and *amoA*-Eubacteria) showed a strong correlation between them (r = 0.83, $p < 0.01$, $n$ = 24) but they did not correlate with genes coding for deni-trification (*nirK* and *nosZ*) that, on the other hand, correlated between each other (r = 0.999, $p < 0.01$, $n$ = 24) (S8 Table). These results suggest that for Soil 8 there was a decoupling of nitrifi-cation and denitrification activity, unlike in Soil 7. In addition, Soil 8 showed a decrease in the number of gene copies along the soil profiles (One-way ANOVA analysis, $p < 0.01$, $n$ = 24) for all five genes analysed as well as reported for Soils 1–6, whereas this trend was not observed for Soil 7.

## Discussion

Nitrogen transformation in soil is a complex process and depends on many factors but there is agreement on the fact that soil microorganisms play a central role [13,17] in regulating the potential for N mineralization, nitrification and denitrification [19].

Results of the study show a strong spatial coincidence (S6 Table) between the numbers of gene copies detected coding for the different N transformations. These results agree with recent indications highlighting the tendency of soil microorganisms to form complex commu-nities within which nitrogen is metabolized, processed and transformed [17]. In addition, gene copies correlate well with mineral nitrogen content (r coefficients > 0.91; $p < 0.05$; $n$ = 252) (S7 Table) for soil that received an ordinary N fertilization (Soils 1–6). Soil nitrate content corre-lated well to gene copies for both soil layers considered (0–50 cm and, above all, 75–100 cm) (S7 Table), while ammonia correlated with gene copies only for the surface layer (0–50 cm) (S7 Table).

### *amoA* Archaea

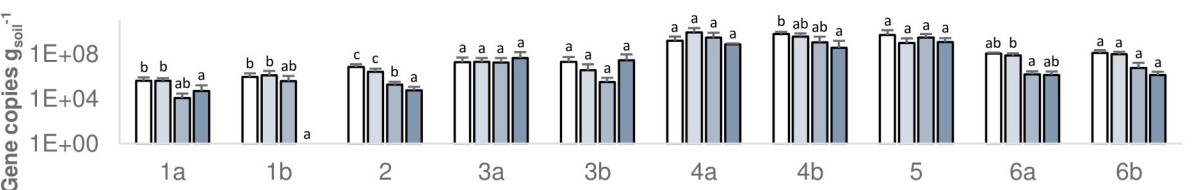

### *amoA* Eubacteria

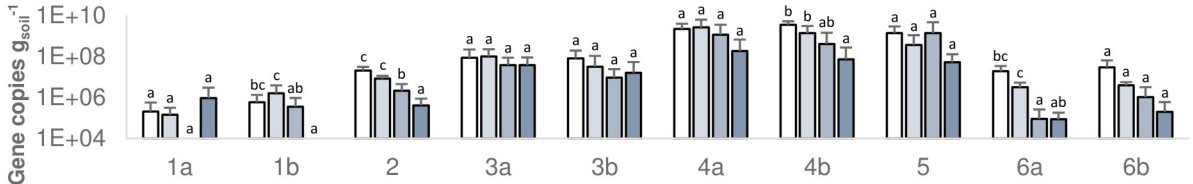

### *nifH*

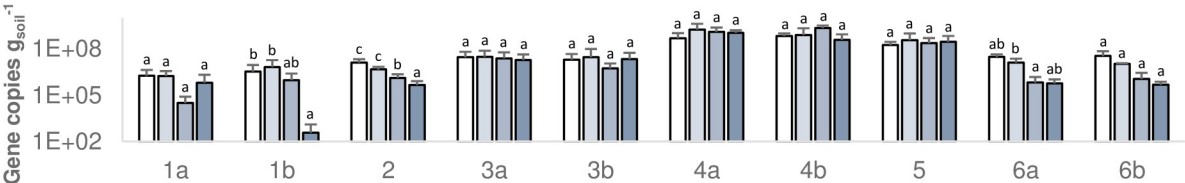

### *nirK*

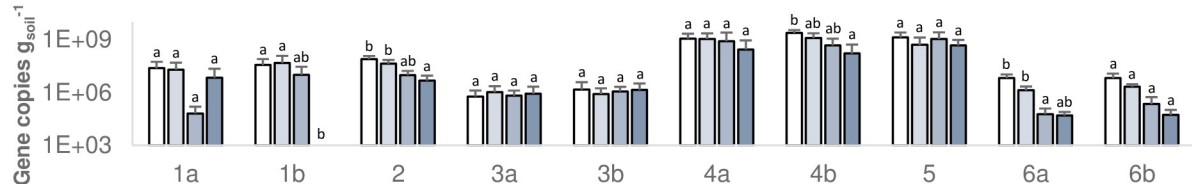

### *nosZ*

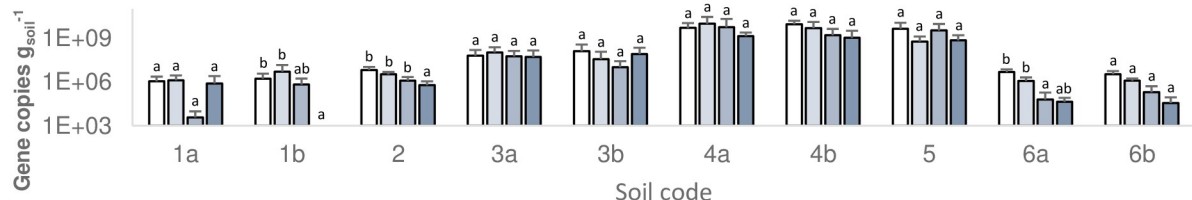

**Fig 3. Gene copies concentration in soil.** Plots show the number of gene copies (average) per gram of soil for each experiment, grouped by sampling depth. The genes analysed are: *amoA* from archaea, *amoA*, *nifH*, *nirK* and *nosZ* from bacteria. Y axis shows $\log_{10}$ scale. Error bars show Standard Deviation, $n = 252$. Letters within the same soil refer to One-way ANOVA tests ($p < 0.05$, Gabriel post-test) performed for each soil and gene.

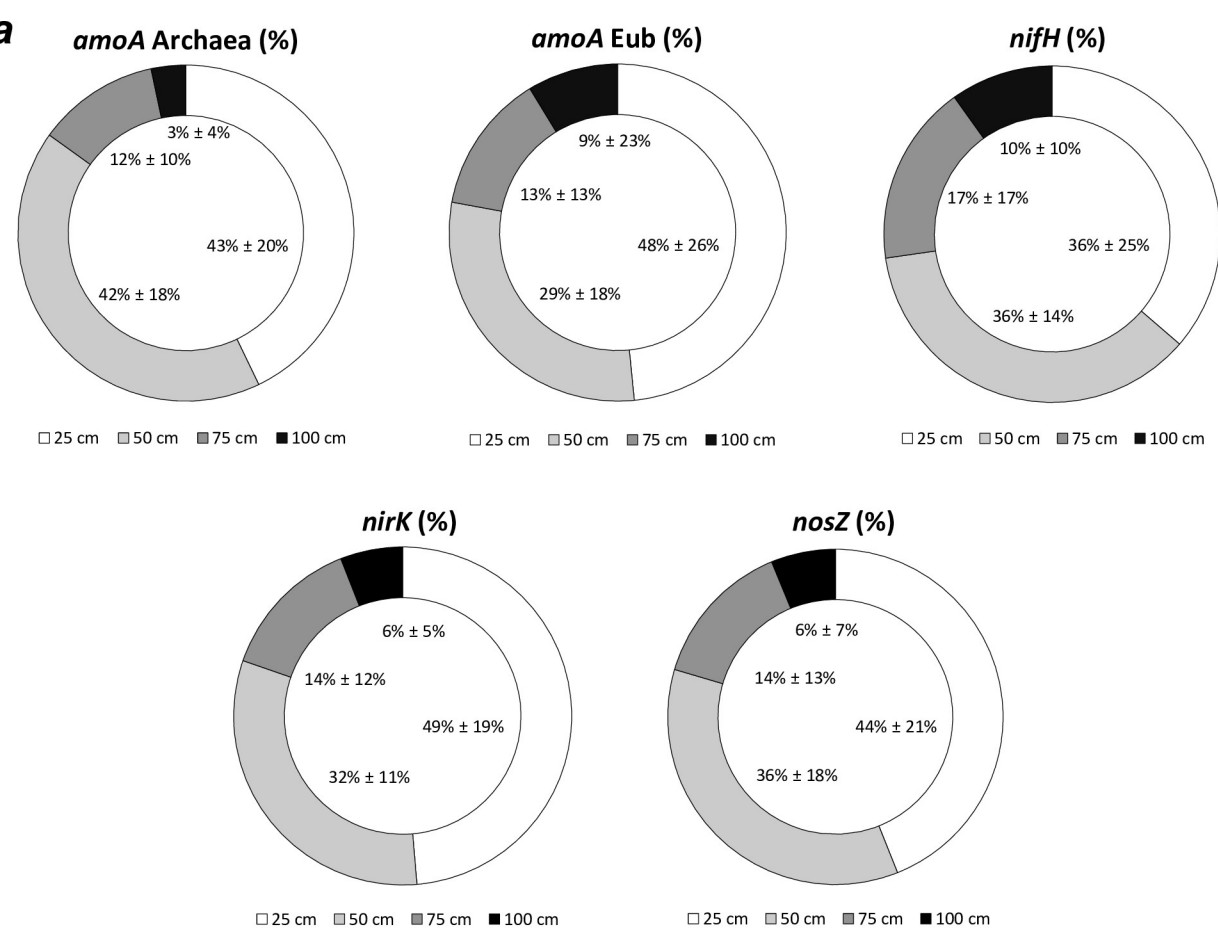

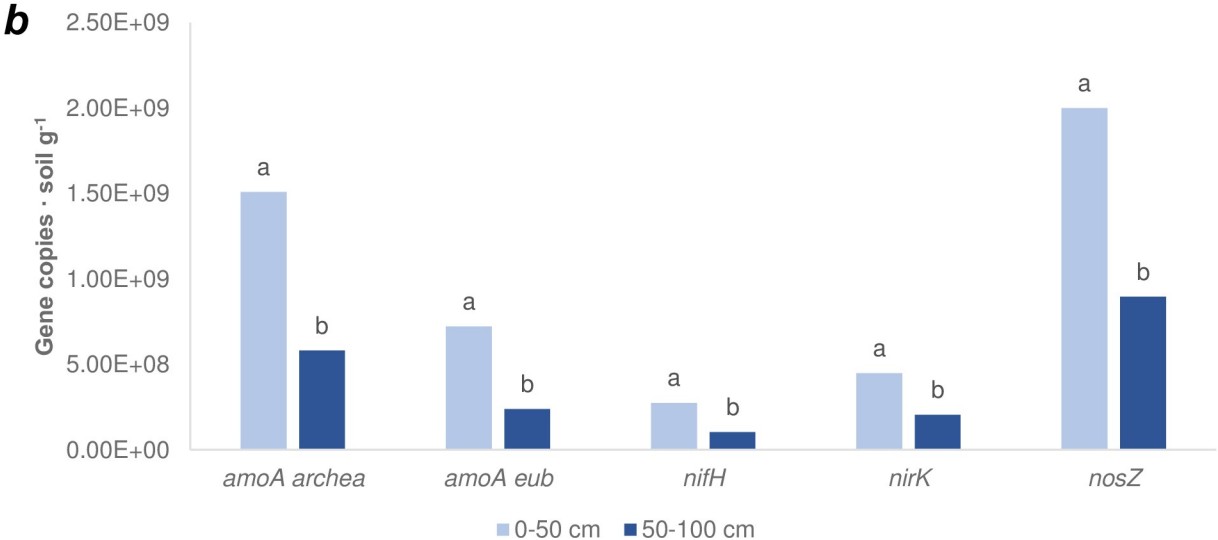

**Fig 4. Microorganisms quantification.** a: percentages of gene copies in all soils analysed, divided into depth classes. Data in the graphic represent the average ± standard deviation of all soils (1a, 1b, 2, 3a, 3b, 4a, 4b, 5, 6a and 6b), (n = 252). b: total number of gene copies (average) per gram of soil, divided into two depth classes (0–50 cm; 50–100 cm), (n = 252). Letters refer to two-way ANOVA analysis (p<0.01, factor 1: depth (two levels), factor 2: sampling period (in correspondence or not to N fertilization) (two levels) (Gabriel test).

Although the abundance of soil microorganisms (i.e. gene copies number) seems to play a fundamental part in determining N speciation in the soils studied, the N dosed, the soil chemical and physical properties, and environmental factors (rainfall) can also play important roles [11,13,17,51,52].

Therefore, we evaluated the contribution of all these factors to N speciation with particular reference to nitrate by multivariate analysis (principal component analysis, PCA). The two-dimensional PCA graph obtained (Fig 5) shows a clear division of soils into three groups, differentiated statistically by $NO_3^-$ concentration at 1-meter depth (two-way ANOVA, p = 0.0003; F = 3.61; DF = 18; $n$ = 59), i.e. Group a, ab and b. Soils of the Group a, in the left quadrants of Fig 5, are characterized very low nitrate concentration at 1-meter depth (1.39 mg $N-NO_3^-$ $kg^{-1}$ ± 1.25; $n$ = 29) (Fig 5). These soils all showed an alkaline pH (8.45 ± 0.03; $n$ = 5) and they are rich in clays and silt (0–50 cm: 28.73% ± 5.07% and 47.14% ± 5.36%, respectively; $n$ = 5) (Table 1). These soils showed a strong reduction of the nitrate concentration from the surface to 1 meter of depth (85% ± 19%; $n$ = 62).

In contrast, the soils of Group b, which occurred in the right quadrants (Fig 5) showed a $NO_3^-$ concentration at a 1-meter depth (8.71 mg $N-NO_3^-$ $kg^{-1}$ ± 4.85; $n$ = 12) significantly greater than that of soils of Group a. The pH measured for these soils is neutral (6.93 ± 0.32; $n$ = 2), and sand is well represented in the first 50 cm (49.95% ± 6.99%; $n$ = 2) (Table 1). The nitrate reduction between the surface layer and the 75–100 cm deep layer was less than that measured for the other soil group but still remarkable, i.e. 58% ± 20% ($n$ = 24). Finally, there is a third group (Group ab) having intermediate values between the other two groups, i.e. $N-NO_3^-$ of 4.80 mg $kg^{-1}$ ± 4.56; $n$ = 18 (Fig 5).

Taking into consideration the different parameters considered in the PCA (Fig 5) it can be seen that, in general, N dose does not affect nitrate presence. The absence of any correlation between the average concentration of $NO_3^-$ detected at 1-meter depth for each soil and the corresponding N supplied as fertilizer (r = 0.26; $p$ = 0.47) confirmed this fact. The soil texture, instead, in particular clay and sand soil contents, seem to affect nitrate concentration along the soil profile (Fig 5), as confirmed by the Pearson correlation analysis for the $NO_3^-$ concentration in the 75–100 cm vs. percentage of sand and clay in the surface layer (0–50 cm), i.e. r = 0.718, $p$<0.05, $n$ = 189 and r = -0.698, $p$<0.05, $n$ = 189, respectively. pH could also play a role (Fig 5) as it is reported that alkaline pH stimulates biological activities [53], although other authors reported the high adaptability of denitrifying bacteria to different pHs [54]. In any case, the reported pH effect contrasts with our results, which indicate that gene copy numbers are higher for those soils characterized by a lower pH (right quadrant of Fig 5) than for soils having alkaline pH (left quadrant of Fig 5). This controversial trend can be explained by considering that gene copies' presence is regulated by the amount of reactive N (r > 0.635, $p$<0.05; $n$ = 189), in agreement with PCA results (Fig 5).

Another point that should be considered consists in the fact that the presence of nitrate at 1 m soil depth depends upon soil water content and so upon rainfall. Data collected for rainfall (Table 2) showed that pluviometry was in line with or even higher than the average rainfall registered for the experimental site, and that nitrate content at 1 m depth did not correlate with rainfall for Soils 1–6 ($p$ = 0.65, $n$ = 57). Therefore, we can exclude the idea that rainfall affected nitrate presence at 1 m depth in normal seasons.

The application of the partial least square analysis (PLS) considering all factors included into PCA analysis gives a regression ($R^2$ = 0.96, $R^2cv$ = 0.95; $p$<0.05; $n$ = 10; $parameters$ = 26) (S9 Table) that confirms all PCA parameters influencing nitrate presence as previously discussed, i.e. genes coding for N transformation and soil texture.

Indeed, high clay and silt contents reduce nitrate concentration in the 75–100 cm soil layer. On the other hand, soils characterized by light textures (sandy soils) are more exposed to

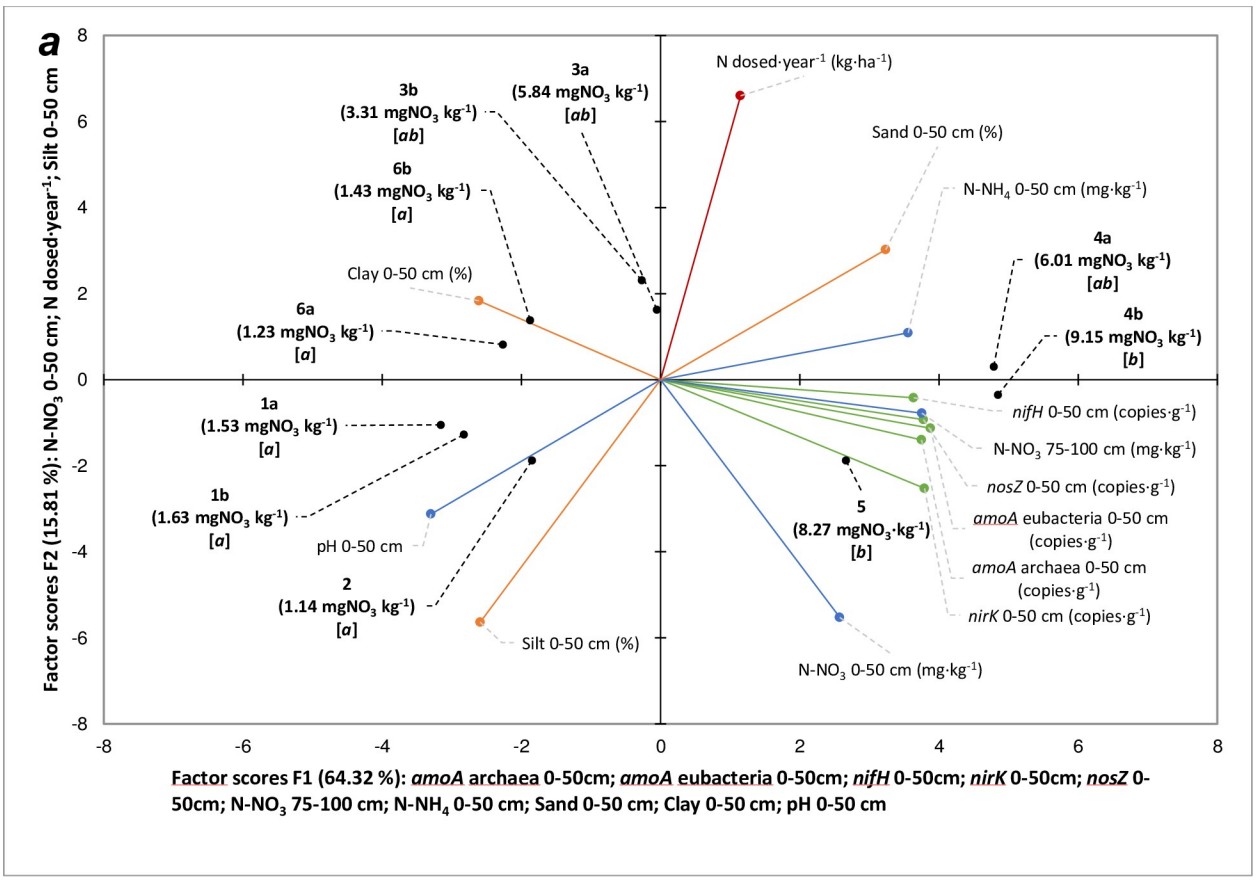

**Fig 5. Principal component analysis (axes F1 and F2: 80.13% of total variance).** a: PCA output. Solid lines show the projections of the initial variables in the factorial space: green lines indicate the number of gene copies in soil, blue lines indicate soil chemical characteristics, orange lines indicate soil physical characteristics, red line indicates the amount of N dosed on the soils. Black dots show the position of the experiments in the factorial space. Round brackets show the average concentration of nitrates measured at 1-meter depth. Square brackets show the result of two-way ANOVA analysis on nitrate concentration ($p < 0.05$; F = 4.1251; Duncan test). Experiments' labels show the soil code. Axis titles report the variables linked to each axe. b: correlation matrix (Pearson) containing: the number of gene copies per gram of soil for *amoA* archaea, *amoA* eubacteria, *nifH*, *nirK* and *nosZ*, the N-NO$_3^-$ and N-NH$_4^+$ concentration (mg kg$^{-1}$ of soil, n = 189. * = the correlation is significant at level 0.05 (two-tailed test)).

nitrate leaching [28] although in this case an increase in the presence of genes related to nitrifying/denitrifying activities (copies of *amoA*-Eubacteria, *amoA*-Archea, *nirK* and *nosZ* genes) driven by the presence of reactive N in the upper layers (S7 Table), is able to keep nitrate concentration in line with that of natural soils (9.6 mg N-NO$_3^-$ kg$^{-1}$)[50], emphasizing the primary role of N-cycle related activities in determining the control of nitrate concentration in soils' profiles.

Therefore, the data discussed indicate that low nitrate presence at 1-meter soil depth in soil ordinarily fertilized (N up to 453 kg Ha$^{-1}$), can be explained by the abundance of gene copies for enzymes related to the N-cycle, but it is clear that soil texture also plays an important role.

Our work considers N transformation within the soil without considering N lost in the air by $NH_3$ volatilization that represents a net loss of nitrogen that does not enter in the soil N cycle. Unfortunately, no direct measurements of ammonia losses were made in this experiment; however, useful data from our previous work performed in the same area, aimed at quantifying ammonia losses for different N fertilizers, indicates N losses as 5.4% (urea) and 18% (digestate spread onto soil) of total N dosed [55]. Therefore, in the future, N losses due to ammonia emission should also be considered in order to quantify correctly total N entering into soil, when making a correct N mass balance. However, $N_2O$ produced during N denitrification, which also represents a net loss going out of the soil system, can be neglected because it was reported to be around 1% of total N dosed [56,57].

However, is there a limit beyond which soils' ability to reduce nitrate presence in 1 m depth soil does end? In order to determine this, two soils (Soil 7 and Soil 8) receiving an excess of N fertilization were considered and studied, and the results reported.

It is interesting to compare the soil $NO_3^-$ behaviour of Soil 7 with Soil 8 in the autumn period because both of these soils received substantial N fertilization which led to different results in terms of nitrate presence at 1 m depth. Indeed, in autumn Soil 7 received a large amount of N (860 kg N Ha$^{-1}$) leading to the high $NO_3^-$ presence in the surface layer, which however did not correspond to high nitrate concentration at the 75–100 cm depth soil layer (Fig 5). On the contrary, in the same period Soil 8 received much less N with similar N-fertilizer (580 kg N Ha$^{-1}$) but showed high nitrate presence at 75–100 cm depth, although rainfall registered was much less than that for Soil 7 (205 mm for Soil 7 and 124 mm for Soil 8) (October-November) (Table 2). As observed for Soils 1–6, rainfall does not correlate with nitrate presence at 1 m depth for Soils 7 and 8 (p = 0.81; $n = 14$).

This result appears more peculiar if we consider that the number of gene copies g$^{-1}$ related to enzymes implicated in nitrifying-denitrifying activities measured for Soil 8 is of 1 to 2 orders lower than that measured for Soil 7 (Fig 3; S5 Table), despite an alkaline pH (Table 1) which could stimulate biological activities [53] and an organic carbon content which can support denitrifying activities (Table 1) [58].

Therefore, both the number of N related gene copies and rainfall events are not consistent with the higher $NO_3^-$ presence at 1-meter depth for Soil 8, indicating that the reason for the difference that occurred between these two soils should be sought in their own properties [28]. Soil 7 contains more clay, especially in the deeper layers (22% of clay in the 0–25 cm and 30% of clay from 25–100 cm) when compared to Soil 8 (from 17.4% in the upper layer to 10.4% in the deeper layer) and we regard this difference as being able to explain the different results on nitrate presence [13]. We can therefore summarize results as follows: Soil 7 received a large amount of ammonia in autumn that was transformed into nitrate by ammonia oxidation microorganisms; then nitrate was concentrated above all in the surface layer, because it was not rapidly leached by rainwater. In this way, the high residential time of ammonia and nitrate due to the abundant clay presence in the soil allowed its denitrification, explaining the low $NO_3^-$ content found at 1-meter depth [59]. This fact was confirmed by both the higher number of gene copies related to nitrifying/denitrifying activities registered for Soil 7 in comparison with Soil 8 (S5 Table), and by the very high positive correlation found for genes coding for nitrification with those for denitrification (*amoA* Archaea vs *nirK*: r = 0.397, $p<0.05$; *amoA* EUB vs *nirK*: r = 0.926, $p<0.01$; $n = 32$; layer 0–100 cm), (*amoA* Archaea vs *nirK*: r = 0.814, $p<0.05$; *amoA* EUB vs *nirK*: r = 0.991, $p<0.01$; $n = 8$; layer 0–25 cm) (S8 Table), indicating that these two processes were strongly linked. The fact that the number of gene copies did not

decrease down the soil profile for Soil 7 (S5 Table) seems to indicate that nitrifying-denitrifying processes continued throughout all soil depths. Contrarily, Soil 8, due to the high proportion of sand, led to the rapid leaching of nitrate, limiting the proliferation of microorganisms (*i.e.* gene copies number) related to N denitrification, that are, in effect, much lower than those measured for Soil 7 (S5 Table).

## Conclusions

Results of this work suggest that with a normal N fertilization (up to 450 kg N Ha$^{-1}$) the microbial populations of the soil involved in the N cycle were able to completely metabolize the nitrogen supplied with fertilization, whatever the soil characteristics, ensuring low nitrate content at one-meter depth. However, for higher N fertilization rates (1,243 kg N Ha$^{-1}$ and 1,470 kg N Ha$^{-1}$), the activity of soil microorganisms was not able to metabolize all the nitrogen. In this case, the characteristics of the soil, i.e. texture, and seasonal rainfall, also regulated the presence of nitrate in soil profiles.

## Supporting information

**S1 Fig.** Ammonium (*a*) and phosphate (*b*) concentration in soil. For each experiment, divided for depth classes, the box plot shows minimum and maximum values (bars), the first and the third quartile (boxes), the median (lines inside boxes) and the average (crosses); *n* = 248. (DOCX)

**S1 Table. Soil studied and soil agronomic management.** In the Table are reported the agronomic data of all the experiments made. (DOCX)

**S2 Table. Sampling periods.** The months of the year in which at least one sampling was carried out for the corresponding Soil are marked in grey. (DOCX)

**S3 Table. Primers used in the quantification of nitrogen related microorganisms in soil.** For each primer are shown name, nucleotide sequence, target gene and reference. N = undefined nucleotide. (DOCX)

**S4 Table. Nitrate, ammonium.** Their concentration (average ± standard deviation) in the analysed soils. (DOCX)

**S5 Table. Gene copies for every microorganism class in the soils analysed.** Average ± standard deviation. (DOCX)

**S6 Table. Pearson correlation matrix between gene copies in regularly fertilized soils.** The test was performed using the data from all the regularly fertilized soils (1–6). The data used refer to the whole profile of the analysed soil (0–100 cm). Numbers in table indicate the r correlation coefficients. (DOCX)

**S7 Table. Pearson correlation matrix.** The test was performed using the data from all the regularly fertilized farms (1–6). Numbers in table indicate the r correlation coefficients. (DOCX)

**S8 Table. Pearson correlation matrix between gene copies in soils fertilized with a N excess.** The test was performed using the data from all the soils 7 and 8. The data used refer to the whole profile of the analysed soil (0–100 cm). Numbers in table indicate the r correlation coefficients.
(DOCX)

**S9 Table. PLS regression.** Regression for nitrate content in soils (75–100 cm) ($n$ = 10) vs. agronomic, chemical, physical, biological and meteorological data (parameters = 26). In the table are reported variables included in regressions and their importance.
(DOCX)

**S1 Graphical abstract.**
(TIF)

## Acknowledgments

CONVENZIONE QUADRO Regione Lombardia–ERSAF, Italy: Supporto tecnico per l'applicazione e il monitoraggio della direttiva nitrati (ARMOSA) + rapporto ambientale per VAS, piano triennale 2014–2016, Agreement Collaborazione tecnico-scientifica per azioni finalizzate a valutare la sostenibilità complessiva della gestione". Project N. 15-3-3014000-218 and 15-3-3014000-219. Gruppo Ricicla lab. Università degli Studi di Milano, Italy, Project N. 1705, 2011: RV_ATT_COM16FADAN_M.

## Author Contributions

**Conceptualization:** Fulvia Tambone, Fabrizio Adani.

**Data curation:** Massimo Zilio, Barbara Scaglia, Andrea Squartini, Fabrizio Adani.

**Formal analysis:** Massimo Zilio, Silvia Motta, Fulvia Tambone, Barbara Scaglia, Andrea Squartini.

**Funding acquisition:** Gabriele Boccasile, Fabrizio Adani.

**Investigation:** Fabrizio Adani.

**Methodology:** Andrea Squartini, Fabrizio Adani.

**Project administration:** Gabriele Boccasile, Fabrizio Adani.

**Software:** Massimo Zilio.

**Supervision:** Fabrizio Adani.

**Validation:** Massimo Zilio, Silvia Motta, Barbara Scaglia, Andrea Squartini, Fabrizio Adani.

**Writing – original draft:** Massimo Zilio, Fabrizio Adani.

**Writing – review & editing:** Massimo Zilio, Fabrizio Adani.

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
