## [Decision Letter · Decision Letter 0]

31 Mar 2020

PONE-D-20-00896

The distribution of functional N-cycle related genes and nitrogen in soil profiles fertilized with mineral and organic N fertilizer

PLOS ONE

Dear Dr. Adani,

Thank you for submitting your manuscript to PLOS ONE. After careful consideration, we feel that it has merit but does not fully meet PLOS ONE’s publication criteria as it currently stands. Therefore, we invite you to submit a revised version of the manuscript that addresses the points raised during the review process.

We would appreciate receiving your revised manuscript by May 29, 2020. To enhance the reproducibility of your results, we recommend that if applicable you deposit your laboratory protocols in protocols.io, where a protocol can be assigned its own identifier (DOI) such that it can be cited independently in the future. For instructions see: http://journals.plos.org/plosone/s/submission-guidelines#loc-laboratory-protocols

We look forward to receiving your revised manuscript.

Kind regards,

Jianlin Shen

Academic Editor

PLOS ONE

Journal Requirements:

3. We note that Figure S1 in your submission contain [map/satellite] images which may be copyrighted. All PLOS content is published under the Creative Commons Attribution License (CC BY 4.0), which means that the manuscript, images, and Supporting Information files will be freely available online, and any third party is permitted to access, download, copy, distribute, and use these materials in any way, even commercially, with proper attribution. For these reasons, we cannot publish previously copyrighted maps or satellite images created using proprietary data, such as Google software (Google Maps, Street View, and Earth). For more information, see our copyright guidelines: http://journals.plos.org/plosone/s/licenses-and-copyright.

a)    You may seek permission from the original copyright holder of Figure S1 to publish the content specifically under the CC BY 4.0 license.  

Reviewers' comments:

Reviewer's Responses to Questions

**Comments to the Author**

1. Is the manuscript technically sound, and do the data support the conclusions?

Reviewer #1: Yes

Reviewer #2: Partly

2. Has the statistical analysis been performed appropriately and rigorously? 

Reviewer #1: No

Reviewer #2: I Don't Know

3. Have the authors made all data underlying the findings in their manuscript fully available?

Reviewer #1: Yes

Reviewer #2: Yes

4. Is the manuscript presented in an intelligible fashion and written in standard English?

Reviewer #1: Yes

Reviewer #2: Yes

5. Review Comments to the Author

Reviewer #1: The manuscript studied the distribution of N cycle related genes in soils with different N fertilizer application and their correlation with soil properties. The topic is interesting and the experimental was also well designed. The major comments to this manuscript are as follows.

L173-177, please simply introduce the methods for soil N and P concentration measurement.

L 180-184, the references for the methods used should be given. Besides, how the soil physical properties were measured?

L 157, the methods for quantifying the nitrogen cycle gene abundances should also be given.

Line 205, the basic properties of soil samples and the climate (e.g. air temperature, precipitation) should be showed in the main text, other than the supporting materials for easy reading.

Line 205, in the Results and Discussion part, some section was written with more results and some section with more discussion. It is better to write results and discussion separately.

Lines 387-393, for deducing the conclusion that when N fertilizer application rate was lower than 450 kg N ha-1, the microbial populations of soil were able to completely metabolize the nitrogen, additional correlation should be done between N fertilizer application rate and soil nitrate content at the deep soil layer.

Fig. 1, 2, 3, 5, the statistical analysis the differences among the sites or different soil layers should be given.

Fig. 4, the PCA analysis should be revised to differentiate soil nitrogen cycle gene abundance and soil chemical and physics properties with different arrows.

Reviewer #2: This manuscript mainly described the importance that N-cycle related genes could reduce Nmin reduction in 1m soil profile under traditional cropland management. However, due to lack the specific data support, at the same time, the author did not consider the N input and output balance, which led to the contribution of N-cycle related genes in reduction NO3 -N leaching existed uncertainty.

In addition, the author did not consider NH3 emission in his research area? The N-cycle related genes did not relate to ammonia emission, so it was questionable about the real contribution of N-cycle related genes about N reduction. At last, the discussion part should add previous research, especially for the N fate, gases emission, 15N trace, and so on to support your conclusion. Based on all of these, a major revision of the MS is recommended before critical commentary on the technical aspects.

Comments for Authors:

General comments:

Title: what did the nitrogen include? Just Nmin or others?

Abstract:

Line 17-20, no related to your research, you should rewrite these sentences.

Introduction

Too much content was about why choose these kinds of genes, vey few previous related research was displayed in this manuscript, which led to no sufficient reasons to support your scientific hypothesis. So, add more related research progress and deleted no related content were necessary.

Such as Line 48-56, no related to your research and suggested to deleted it

Line 125, what’s chemical data mean?

Material and Methods

This part should add the detailed cropland management (such as irrigation, fertilizer amount, times, cropping system, tillage and so on) and climatic conditions.

Line 133, add more detailed information about why choose these sites

Line 183, add references to support your analysis methods.

Results and Discussion

The discussion did not write well, It was not enough to illustrate that the N-cycle related genes could reduce NO3-N by just PCR correlation results. You should add the data about the contribution amount from N-cycle related genes microorganism. In addition, a lot of studies demonstrated that the NO3-N could leach more than 1m depth and even into underground water under traditional farm management in long-term experiment. So, it was worth to consider whether large amount NO3-N still existed in soil profile due to low soil water content, and if high precipitation could lead to N leaching into deeper soil layer.

Line 234-246. Redundant content

Line 266, what is group a, group b, group ab? Too much factors were displayed in Fig 4, which led to difficult to understand your mean. So ,I suggest to improve this fig, or divide into several figs.

6. PLOS authors have the option to publish the peer review history of their article (what does this mean?). If published, this will include your full peer review and any attached files.

Reviewer #1: No

Reviewer #2: No

---

## [Author Response · Author response to Decision Letter 0]

1 May 2020

5. Review Comments to the Author

Reviewer #1: The manuscript studied the distribution of N cycle related genes in soils with different N fertilizer application and their correlation with soil properties. The topic is interesting and the experimental was also well designed. The major comments to this manuscript are as follows.

L173-177, please simply introduce the methods for soil N and P concentration measurement.

WE: Ok, we have better described the method used, indicating the publications to which we have referred (see marked version).

L 180-184, the references for the methods used should be given. Besides, how the soil physical properties were measured?

WE: We put together all the soil analysis in a unique paragraph “Soil analysis”, and we have entered all references to analytical methods. For soil physical properties we mean texture, now “physical properties” have been replaced with “texture” (see marked version).

L 157, the methods for quantifying the nitrogen cycle gene abundances should also be given.

WE: Paragraph has been revised (see marked version)

Line 205, the basic properties of soil samples and the climate (e.g. air temperature, precipitation) should be showed in the main text, other than the supporting materials for easy reading.

WE: Ok, Table S4 (soil characteristics) and Table S9 (rainfall) are now moved into the main text, becoming Table 1 and Table 2. Moreover, rainfall Table has been amended reporting, also, average data for 2014-2019 to be compared with rainfall measured during experiment (such as requested also by Referee 2) (see marked version), to better support both result and discussion.

Line 205, in the Results and Discussion part, some section was written with more results and some section with more discussion. It is better to write results and discussion separately.

WE: We have re-arranged the text dividing result and discussion so that now paper is clearer. Moreover, many additions have been made to better explain both results and to make discussion, taking into consideration, also, Referee 2 suggestions (see marked version). 

Lines 387-393, for deducing the conclusion that when N fertilizer application rate was lower than 450 kg N ha-1, the microbial populations of soil were able to completely metabolize the nitrogen, additional correlation should be done between N fertilizer application rate and soil nitrate content at the deep soil layer.

WE: ok, following the advice we performed a linear regression analysis between the amount of N dosed on the soils during the year and the average nitrate concentration measured at 1-meter depth in the same soils. The analysis have been discussed in the main text (see marked version). 

Fig. 1, 2, 3, 5, the statistical analysis the differences among the sites or different soil layers should be given.

WE: As suggested by the reviewer, we added a statistical analysis to Figures 1,2, and 5. Figures 3 and 4 already presented a statistical analysis, so they have not been changed

Fig. 4, the PCA analysis should be revised to differentiate soil nitrogen cycle gene abundance and soil chemical and physics properties with different arrows.

WE: ok, we have modified Figure 4, differentiating the lines that refer to different types of variables using different colours.

Reviewer #2: This manuscript mainly described the importance that N-cycle related genes could reduce Nmin reduction in 1m soil profile under traditional cropland management. However, due to lack the specific data support, at the same time, the author did not consider the N input and output balance, which led to the contribution of N-cycle related genes in reduction NO3 -N leaching existed uncertainty.

In addition, the author did not consider NH3 emission in his research area? The N-cycle related genes did not relate to ammonia emission, so it was questionable about the real contribution of N-cycle related genes about N reduction. At last, the discussion part should add previous research, especially for the N fate, gases emission, 15N trace, and so on to support your conclusion. Based on all of these, a major revision of the MS is recommended before critical commentary on the technical aspects.

WE: thank you to referee; He/She is right reporting that also N can be lost and should be considered into N balance. Therefore, we have added a paragraph about this in the Discussion section (see marked version) adding also the literature. 

In addition, Result and Discussion have been now divided (such as requested also by Referee 1) so that paper is clearer (we hope so). 

Comments for Authors:

General comments:

Title: what did the nitrogen include? Just Nmin or others?

We: we added “ammonia and nitrate nitrogen”

Abstract:

Line 17-20, no related to your research, you should rewrite these sentences.

WE: we deleted this part. 

Introduction

Too much content was about why choose these kinds of genes, vey few previous related research was displayed in this manuscript, which led to no sufficient reasons to support your scientific hypothesis. So, add more related research progress and deleted no related content were necessary.

We: Introduction has been revised and referee suggestions considered to improve the introduction part (see marked version). 

Such as Line 48-56, no related to your research and suggested to deleted it

WE: we deleted this part according to referee’s suggestion (see marked version)

Line 125, what’s chemical data mean?

We: we have explained what it does mean (see marked version).

Material and Methods

This part should add the detailed cropland management (such as irrigation, fertilizer amount, times, cropping system, tillage and so on) and climatic conditions.

WE: OK, S1 Table now reports: Cropland management, fertilizers used, time of fertilization, total N supplied, crop harvest date, land extension, presence of irrigation and tillage vs. no tillage (see marked version). 

Line 133, add more detailed information about why choose these sites

WE: These information have been added such as also requested by referee 1 in the Experimental site chapter (see marked version).

Line 183, add references to support your analysis methods.

WE: OK. References have been added such as requested, also, by referee 1 in Soil Analysis section (see marked version).

Results and Discussion

The discussion did not write well, It was not enough to illustrate that the N-cycle related genes could reduce NO3-N by just PCR correlation results. You should add the data about the contribution amount from N-cycle related genes microorganism. 

WE: we have re-arranged the text (such requested also by referee 1) dividing Result and Discussion sections to make test more understandable. Now it is clear (from our point of view) that microorganisms play an important role in N transformation, although, also soil characteristics (soil texture) play a role (see marked version, both Result and Discussion sections).

Moreover, Referee asked for studying the contribution of microorganism. Unfortunately, we did not plan to do it so that we cannot satisfy the referee. Anyway, we have chosen gene copies approach, as it is able resuming and quantifying gene presence related to N-cycle such as proposed previously by other Authors: e.g.

Kuypers MMM, Marchant HK, Kartal B. The microbial nitrogen-cycling network. Nat Rev Microbiol. 2018;16: 263–276. doi:10.1038/nrmicro.2018.9

Sanford RA, Wagner DD, Wu Q, Chee-Sanford JC, Thomas SH, Cruz-Garcia C, et al. Unexpected nondenitrifier nitrous oxide reductase gene diversity and abundance in soils. Proc Natl Acad Sci. 2012;109: 19709–19714. doi:10.1073/pnas.1211238109

Leininger, S. et al. Archaea predominate among ammonia-oxidizing prokaryotes in soils. Nature 442, 806–809 (2006).

Rotthauwe, J. H., Witzel, K. P. & Liesack, W. The ammonia monooxygenase structural gene amoA as a functional marker: molecular fine-scale analysis of natural ammonia-oxidizing populations. Appl. Environ. Microbiol. 63, 4704–12 (1997).

Feld, L. et al. Pesticide side effects in an agricultural soil ecosystem as measured by amoA expression quantification and bacterial diversity changes. PLoS One 10, 1–20 (2015).

In addition, a lot of studies demonstrated that the NO3-N could leach more than 1m depth and even into underground water under traditional farm management in long-term experiment. So, it was worth to consider whether large amount NO3-N still existed in soil profile due to low soil water content, and if high precipitation could lead to N leaching into deeper soil layer.

WE: Referee is right. Maybe this point was not very clear because rainfall data were reported in the Supporting information. New paper version includes in the text rainfall registered for the season considered and average rainfall data for a proximate periods (2014-2019) to better explain results discussed in both Results and Discussion sections (see marked text for additions). Anyway, rainfall has been considered in the PCA analysis and it was not resulted affecting nitrate presence at 1 m depth differently by other parameters. 

Line 234-246. Redundant content

WE: This part has been partially removed and partially re-arranged. Because now Result and Discussion sections are two separate sections please refers to Result - subsection: The abundance of gene copies related to the N-cycle (marked version).

Line 266, what is group a, group b, group ab? Too much factors were displayed in Fig 4, which led to difficult to understand your mean. So ,I suggest to improve this fig, or divide into several figs.

WE: colouring arrows have improved Figure 4. Doing so now the Figure is easier to be understand. In addition, in the text what group a, ab and b are, has been explained (see marked version)

---

## [Editor Report · Decision Letter 1]

13 May 2020

The distribution of functional N-cycle related genes and ammonia and nitrate nitrogen in soil profiles fertilized with mineral and organic N fertilizer

PONE-D-20-00896R1

Dear Dr. Adani,

We are pleased to inform you that your manuscript has been judged scientifically suitable for publication and will be formally accepted for publication once it complies with all outstanding technical requirements.

With kind regards,

Jianlin Shen

Academic Editor

PLOS ONE
---

## [Editor Report · Acceptance letter]

22 May 2020

PONE-D-20-00896R1 

The distribution of functional N-cycle related genes and ammonia and nitrate nitrogen in soil profiles fertilized with mineral and organic N fertilizer 

Dear Dr. Adani:

I am pleased to inform you that your manuscript has been deemed suitable for publication in PLOS ONE. Congratulations! Your manuscript is now with our production department. 

With kind regards,

on behalf of

Dr. Jianlin Shen 

Academic Editor

PLOS ONE